# Building H.O.U.S.E (Healthy Outcomes Using a Supportive Environment): Exploring the Role of Affordable and Inclusive Housing for LGBTQIA+ Older Adults

**DOI:** 10.3390/ijerph19031699

**Published:** 2022-02-01

**Authors:** Amy Rosenwohl-Mack, Darin Smith, Meredith Greene, Karyn Skultety, Madeline Deutsch, Leslie Dubbin, Jason D. Flatt

**Affiliations:** 1Department of Social and Behavioral Sciences, School of Nursing, University of California San Francisco, San Francisco, CA 94143, USA; amy.rosenwohl-mack@ucsf.edu (A.R.-M.); leslie.dubbin@ucsf.edu (L.D.); 2Office of Community Engagement & Support, Oregon Health and Science University, Portland, OR 97239, USA; smidar@ohsu.edu; 3Division of Geriatrics, Department of Medicine, School of Medicine, University of California San Francisco, San Francisco, CA 94143, USA; meredith.greene@ucsf.edu; 4Openhouse, San Francisco, CA 94102, USA; karynskultety@gmail.com; 5Department of Family and Community Medicine, School of Medicine, University of California San Francisco, San Francisco, CA 94143, USA; madeline.deutsch@ucsf.edu; 6Department of Social and Behavioral Health Sciences, School of Public Health, University of Nevada, Las Vegas, Las Vegas, NV 89154, USA

**Keywords:** housing, LGBTQIA+, older adults, HIV/AIDS, inclusion, social support

## Abstract

Little is known about how permanent, inclusive, affordable, and supportive long-term housing may affect the health of low-income lesbian, gay, bisexual, transgender, queer, intersex, asexual and/or another identity (LGBTQIA+) older adults. Focus group interviews were conducted with 21 older adults to explore the lived experiences and potential health benefits of living in a new LGBTQIA+-welcoming senior housing. Participants reported that moving into the housing was associated with benefits for health and well-being, especially for psychological health. Community, social support, and in-house services were particularly important. However, the combined nature of LGBTQIA+-welcoming and older adult only housing evoked mixed feelings. Appropriate and accessible housing solutions are essential for LGBTQIA+ older adults and may help address health disparities for these populations.

## 1. Introduction

Older adults who identify as lesbian, gay, bisexual, transgender, queer, intersex, asexual and additional identities (LGBTQIA+) experience significant health disparities compared to heterosexual cisgender (i.e., not transgender) counterparts. They report higher rates of disability and poor physical and mental health, and they are less likely to be partnered or married [1]. Loneliness is also common for LGBTQIA+ older adults living with HIV/AIDS [2]. LGBTQIA+ older adults report critical needs for housing, transportation, and social support, but few aging service providers offer LGBTQIA+-tailored services [1,3]. In addition to needing these focused health and aging services, LGBTQIA+ older adults face significant challenges in obtaining affordable, welcoming, and supportive long-term housing, which may exacerbate their health and aging concerns. These challenges are due to the lack of affordable or subsidized housing, discrimination, and financial hardship, among other factors [4].

It has been suggested that LGBTQIA+ older adults would likely need to rely on senior housing earlier than their cisgender, heterosexual counterparts, given the need for inclusive and supportive environments and fears of discrimination in long-term care settings [5,6,7]. A qualitative study conducted with LGBTQIA+ older adults living in LGBTQIA+-friendly housing communities found that social relationships and acceptance were two important resources for these residents [5]. Fears of “returning to the closet” were also mentioned if one had to move to less inclusive housing. Studies with LGBTQIA+ older adults in urban cities have identified housing assistance as one of the top four most needed services [8,9,10,11]. While most LGBTQIA+ older adults lived in an apartment or house, more than 5% were living in single occupancy rooms (SROs), residential hotels or were homeless.

LGBTQIA+ older adults are also more likely to rent their homes (54%), while 28% owned their home but were still making mortgage payments, and only 13% owned their homes outright; 5% reported having to rely on financial support for housing. Nearly 70% of LGBTQIA+ older adults reported low confidence in their ability to stay in current housing [8,10,11]. This was due to socioeconomic factors including risk of foreclosure (54%), health challenges (44%), and aging-related accommodation needs (40%), such as installation of safety/grab bars or elevators. Housing instability has also been found to be higher among LGBTQIA+ older adults who live alone and have lower incomes and education attainment. Finally, nearly 50% of LGBTQIA+ older adults reported feeling unsafe in obtaining housing assistance due to past experiences with discrimination [8,9,10,11].

LGBTQIA+ older adults are significantly burdened by health conditions compared to their cisgender, heterosexual peers, including poor general health, physical disabilities, chronic conditions, including living with HIV/AIDS (up to 25%), and greater mental health distress and depression [12,13]. LGBTQIA+ older adults also experience greater behavioral and psychosocial risks, including heavy smoking, obesity and alcohol use [13,14,15,16,17] and depression [12,18,19,20]. Many of these health disparities are linked to historical and everyday experiences of discrimination [21]. Examining social challenges for LGBTQIA+ older adults, more than a third live alone, and 3 out of 5 are neither partnered nor married [20,22]. In addition, only 15% of LGBTQIA+ older adults report having children and of these, 60% reported that their children would not be able to help them with caregiving needs [1,6,23,24,25,26]. These challenges in accessing social and care challenges are often exacerbated by economic challenges. More than 40% reported not having the minimum income necessary to meet their basic needs (based on Elder Economic Security Index), and 30% reported an income of 200% below the federal poverty level [8,11]. Finally, nearly 50% reported experiencing discrimination in the past year, with 81% of transgender older adults reporting discrimination in the past year [8]. However, potential resilience factors should also be considered for LGBTQIA+ older adults, such as community connections, unique social networks including chosen family and friends, and greater engagement in social, leisure and wellness activities [1]. These factors may also be important to linkages to housing and related health resources.

Links between housing and health status for non-LGBTQIA+ populations are well established. Experiencing housing disadvantage (including precarity and poor physical quality) has negative effects on future mental health [27], and poor-quality housing is associated with worse health outcomes for those living with HIV/AIDS [28]. In addition, housing-related interventions and improvements at both the individual and community level can improve health, including respiratory function, quality of life, and mental health [29,30,31]. Housing First models, which provide housing as a first-line, low-threshold intervention for chronically homeless people, continue to provide robust data on the effectiveness of housing as a health intervention [31].

However, there is limited data on the relationship between housing and health for LGBTQIA+ older adults. Housing is frequently reported as a concern in studies on the needs of LGBTQ aging populations [6,32,33,34,35], and LGBTQIA+ older adults may experience discrimination when seeking to rent or buy housing [4]. Home ownership appears to be lower among LGBTQIA+ people, who are also more likely to face socioeconomic challenges than their heterosexual cisgender counterparts [36]. In addition, LGBTQIA+ older adults fear discrimination from aging and long-term care providers [35,37]. A national survey found more than 75% would not disclose their LGBTQIA+ identity if they had to live in elder care settings (Justice in Aging, 2015).

Little is currently known about how permanent, inclusive, affordable, and supportive long-term housing environments may affect the health of low-income LGBTQIA+ older adults. Only one published study reports on the experiences of older adults living in LGBT senior living communities [5]. Major themes included safety, community support, diversity and inclusivity. However, the study sites were not affordable housing and appear to have been privately owned, suggesting that they were likely inaccessible to many low-income LGBTQIA+ older adults and may not reflect wider experiences. A study of an LGBTQIA+-tailored recovery housing for men of all ages found that the effects of discrimination were reduced and that the setting helped to create positive social networks, highlighting the potential value of LGBTQIA+-welcoming spaces for health [38]. In light of the limited existing literature, Portacolone and Halpern (2016) [39] called for more research on experiences of those who move to “age- and sexual-orientation-segregated” housing. In this study, we explored the lived experiences of older adults currently residing in a new LGBTQIA+-welcoming and affordable senior housing building, with a focus on perceived benefits for health and well-being.

## 2. Materials and Methods

We explored experiences of older adults living in an LGBTQIA+-welcoming and affordable senior housing building in a Western U.S. Metropolitan area by conducting a series of three focus groups from July to August 2018. It is estimated that between 18,000 and 20,000 LGBTQIA+ older adults aged 60 years or older currently reside in this Western U.S. Metropolitan area [7]. These groups formed part of a larger longitudinal study on the health effects of living in LGBTQIA+-welcoming and affordable senior housing. The study was approved by the Institutional Review Board at the University of California, San Francisco.

The setting was an LGBTQIA+-welcoming and affordable senior housing development that opened in 2017, after almost ten years of planning and negotiation. We use the pseudonym WESTHOME throughout this manuscript. It was developed as a partnership between a local nonprofit and a national affordable housing provider, with support from the local city administration. This redevelopment of an historic site retained many esthetic features of the original building, which was previously a school. The housing site includes ten studio apartments, 26 one-bedroom apartments, and four two-bedroom apartments, with eight units set aside for people who were previously homeless and living with HIV/AIDS. Monthly rents ranged from $800–$1200 (considered affordable for the region). The units were allocated through the local city government’s lottery system, in which people aged 55 and older with an income not exceeding 50% of Area Median Income (around $39,000 for a one-person household) were eligible to apply; over 1800 people entered. People of any gender identity and sexual orientation were eligible, and over two-thirds of those who moved in identified as LGBTQIA+, likely due to the local LGBTQIA+ aging nonprofit’s advocacy for community members to apply. The housing is co-located with an LGBTQIA+ senior center and features on-site case management and social activities as well as regular community events.

The theoretical framework for this study was “health equity promotion”, which provides a life course perspective on how historical and social factors have influenced the health of LGBTQIA+ older adults [40]. We chose to conduct focus groups to collect a range of perspectives on our research questions, to identify areas of consensus and disagreements, and to gain insight into social relationships between members of the community. Any current resident of the housing was eligible to participate. Recruitment procedures included distributing flyers within the building and outreach by staff from the nonprofit co-located within the housing complex. Twenty-six participants were initially recruited to participate and contacted by phone to schedule the date of the focus group interview. Each participant was given a $25 grocery store gift card, and a meal was provided at each focus group. The groups were held in community rooms within the complex, and each lasted 75–93 min.

Focus group procedures involved two facilitators being present for each focus group; one led the group and the other took notes on dynamics that might not be discernible from the audio recording alone. At the beginning of each focus group, the facilitator presented information on informed consent, including risks, benefits, and confidentiality, inviting participants to ask questions before they consented verbally to participate. We also asked participants to complete a demographic survey. Our semi-structured focus group guide was developed by the authors and included questions on: (1) how participants felt when they heard they would be moving to the housing site; (2) what they liked most about the housing site; (3) how comfortable they felt being open about their sexual and/or gender identity in their current living situation; (4) health challenges; (5) changes in health since moving to the housing site; and (6) current service utilization. Additional probes prompted participants to reflect on their past experiences and potential future health and aging needs, as well as encouraging them to consider health in a broad sense, encompassing social and psychological well-being as well as physical conditions. Each focus group was audio recorded and transcribed verbatim.

For data analysis, we used a “flexible coding” approach adapted from the analytic process described by Deterding and Waters [41]. After reading the transcripts and listening to the recordings to familiarize ourselves with the content and participants, our first step was to “index” the transcripts, connecting the content to the questions in the focus group guide. We then focused more closely on the experiences and processes described in response to each question, applying analytic codes. We produced memos focusing on each participant separately, collating their contributions to the discussion and forming a picture of their individual trajectories, as well as memos about each focus group as a whole. This enabled us to balance abstraction across cases with representation of the diverse personal experiences of the participants in our study. Through weekly analytic meetings we compared our codes, memos, and emerging interpretations, testing our conceptual accounts against the data from the three focus groups and identifying points of agreement and divergence. The themes described in this paper reflect the results of our iterative and collaborative interpretive process.

## 3. Results

We recruited 21 participants, with seven in the first group, nine in the second, and five in the final focus group; Table 1 summarizes their demographic characteristics.

### 3.1. Qualitative Findings

#### 3.1.1. Housing Stability Improves Mental Health and Reduces Stress

Most participants described living in unstable or unsuitable housing prior to moving into WESTHOME. Their experiences indicated a range of stressful housing situations that LGBTQIA+ older adults may face. Some had been previously homeless, and others were in temporary or precarious housing. Many described improvements in their health and well-being now after moving to a more stable situation at WESTHOME.

A Mexican American transgender woman in her late 50s had been living temporarily in an expensive apartment with two other transgender women before moving to WESTHOME; she described that arrangement as “buying time, in essence”. Moving to stable, subsidized housing was a great relief for her, meaning that she no longer had to worry about money and now had more time to care for herself:
*My mental well-being seems to be healthier, [and] my physical [well-being]. Financially, it’s affordable for me, so that relieves a lot of the stress and stuff. Because coming up with the rent that we had to come up with before was like … exhausting. So we get to kind of enjoy a little bit more free time, in essence, for me personally. My health seems to be relatively good.*

The stability of living at WESTHOME also enabled residents to address other stressors in their lives. A Black, straight, transgender woman in her early 60s was living in one of the units reserved for those who had experienced homelessness. Prior to moving to WESTHOME she lived in a single room occupancy where she did not always feel safe. As soon as she saw WESTHOME, she recalled thinking: “I knew I was gonna stay here”. She described how she now felt safe and respected; part of this was living in a well-lit area with vigilant neighbors, but she also felt more control over her environment:
*It helps my confidence. I don’t bring [over] people that I know don’t deserve it [even if] I sometimes desire to hang with. I can go to their neck of the woods. But they’re not invited to mine. I’m not saying I’m better than anybody around this area but this is like a safe, honest-system place.*

WESTHOME offered both increased opportunities for security compared to her previous housing and a sense that she deserved to live in a positive and safe space, where she could protect herself from negative influences.

For a white gay cisgender man in his mid 60s who used a wheelchair, moving to WESTHOME represented a huge change. He had been living in an assisted living facility for 20+ years, including five years in hospice care. In the assisted living, he experienced homophobia and felt that he did not need the level of care provided. Moving after so long was a major upheaval, but he said: “I thought to myself, I been through so much in the past 25 years that if I don’t make this move, I’m never going to get another chance. And I just went for it”. He described how he now felt each morning:
*I wake up and I say, “Oh, did I dream that or is it real.” And I get up and I realize, man, I’ve died and gone to heaven. This is perfect. I have a beautiful, one-bedroom garden apartment. I love that. It’s just … I just feel so at home.*

Like many other participants, the relief of having stable housing played a significant role in his improved well-being: “It’s hard to describe, but it really is an uplifting thing. Because number one, you have the weight of trying to find housing off your shoulders. […] And once that goes, anything is possible”. His own trajectory exemplified this: “I went from hospice care, to assisted living, I now live independently. I went back to work and I’m not stopping”. These exemplars demonstrate the extent of the health benefits associated with reduced housing stress for new residents at WESTHOME.

#### 3.1.2. Physical Environment Promotes Well-Being and Healing

In addition to the general sense of stability WESTHOME provided, residents also felt pride in their home environment. Some participants reported issues with the move-in process, but once they were in, they described having housing at WESTHOME as “a gift from the universe” and “the greatest prize that I could have”. In every focus group, participants praised the physical appearance of the building, describing it as “historic”, “beautiful”, and a “terrific space”; one resident summarized: “You upgraded”. Others appreciated the location and quiet space, the light, and opportunities to exercise by walking around the building, while the elevator was reassuring to a participant who was having surgery soon.

An American Indian/white straight trans woman in her 50s had been selected to live at WESTHOME through a homeless support program. She described having loved her old apartment, but moving to WESTHOME was a whole new experience: “Even just as I walk in my entry way, I get this sense of “wow”, I get this whole like “wow” moment”. The layout and feel of the building were especially important to her:
*The architecture… it’s round… so energy flows really amazingly here, even just walking down the hallway, I get a sense of relief, whereas I’m walking in other apartment buildings, visiting friends, the hallways are about as wide as this [arm length indicated a narrow space], it seems very sterile.*

The building design also contributed to her health: “When I am in a space where my positive energy can flow I think my health is way better. Mentally, physically, spiritually. All of it”. After recent surgeries, she recognized how being in a positive space, both in terms of the physical environment and the community around, had helped her to recover: “I really believe my mind can heal it all, but if you’re in a space where that’s not able to flourish, your body suffers”.

Other participants focused on more tangible aspects of the building. A Black heterosexual cisgender woman in her late 50s had been renting an apartment in a rent-controlled building that had since been sold, and she was anxious about what would happen to her. After moving to WESTHOME, she felt much more relaxed: “I have really good neighbors. I feel comfortable. I feel safe. In the building, I feel safe”. She had experienced some physical health issues since moving in but felt better mentally without the anxiety associated with her previous housing. Reflecting on an upcoming surgery, she felt that living at WESTHOME with all the facilities available to her would help her heal:
*I’m glad I’m in this building because there’s an elevator. I don’t have to take any stairs. Because I’m on the third floor, I can walk. And I think I can heal faster because I’m in this building.*

A Latinx/white gay cisgender man in his early sixties, who was an AIDS survivor, had been living in a historically gay neighborhood since the early 90s but did not feel much of a sense of security, worrying about the high cost of rent: “In this town, if you lose your place, it’s goodbye”. He described his tiny apartment and stressful experiences with neighbors as a “hell situation that I was stuck in for eight years”. The contrast he described after moving to WESTHOME was striking, in terms of both emotional well-being and access to amenities: “That sense of relief […] having a full-size refrigerator, a bathroom, a bathtub, and stuff like that, and not fearing every time you went out of your room”. The unique feel of the building was important for his well-being: “it’s a beautiful building. It’s not like the cookie cutters we see going up the city, which is really kind of horrifying”. Tracing his feelings since moving in, he described having felt “really small in these giant hallways” at first but was now feeling “bigger”—taking up more space as his physical and emotional health improved.

Several participants had been familiar with the building in its previous incarnations, and one had even designed some of the original décor. Personal connections with the building’s history made it especially attractive. One resident, a white gay cisgender man, had taken classes in the building previously. The high ceilings allowed him to install plenty of storage cabinets in his apartment, and the amount of natural light felt special: “You just don’t find that in a modern building. I walk up here and see these poor people paying $3000, $4000, $5000 a month for these little boxes! And I think, ‘Jesus, how did I fall into this?’” He was also very aware of how rare it was to be able to stay in the city:
*So many of my friends had simply been driven out of the city by the levels of rent that they had to pay. Ended up in Florida and other ghastly places. And here I’m allowed to stay. It’s wonderful.*

He had chosen a studio apartment, which he found less stressful: “To me, living in a smaller space relieves a lot of good pressure, housekeeping all the time. I’ve got time to work now. I don’t have to fuss with dusting every five minutes”.

#### 3.1.3. Community and Social Support Are Protective for Health

##### Creating the Community

The combination of the well-designed physical space and the strong social bonds they had created helped residents feel supported and included. They saw themselves as having actively “created” rather than simply “joined” a community, a feeling that was perhaps encouraged by the founder of WESTHOME who was reported to have said: “We’re giving this group of people this situation, to take it and run, and develop it into whatever kind of community they want it to be […] that’s never going to happen again so take that opportunity and run with it”. Several participants pointed to the fact that they had a “common bond of all coming at the same time” and “being part of the beginning”, which generated mutual respect and a lack of hierarchy. The sense of community and harmony among the residents was seen as particularly impressive given their diverse backgrounds and experiences:
*The unity in this building is phenomenal. We’re all different walks of life, different ethnicities, but when we all come together, we’re all one. […] Everybody supports everybody. We look out for each other and we make sure that everybody is well taken care of and that’s rare. (African American straight cisgender woman)*

##### Supporting Each Other

Several residents were surprised to realize how much they valued the sense of community and presence of social support at WESTHOME. A white asexual/bisexual man in his late 50s had been living with his brother for nearly 30 years, but the situation was becoming untenable due to his brother’s increasingly severe issues with hoarding. He was allocated one of the units reserved for people who had previously been homeless. He described how he had not realized how isolated he had become until he moved to WESTHOME:
*Once I got here, everybody was saying, “Welcome to the community”, and all this stuff. I was like, what is this whole community thing? I had no idea that this place had even existed. I realized that I really needed that. That I needed a supportive environment because I was starting to feel isolated, getting older. So for me it’s been a really positive experience.*

He reported feeling better both physically and emotionally, as well as being able to take better care of himself:
*My mood has gotten much better and my health has improved. I wake up with a smile on my face, I don’t feel isolated. I’m just taking much better care of myself, so it’s really helped a lot. And if you do have a bad day, you can talk to a social worker, you can talk to somebody else in the building. There’s always that support system here. So that helps a lot with not feeling isolated.*

A white gay cisgender man in his early 70s had been living in a rent-controlled unit, but his landlord was in his 90s and he worried that when the landlord died, his children would move in and force him out. Since moving, he had noticed significant changes in his social life and sense of self. Describing himself as “not really a joiner […] kind of private”, he had nonetheless become a regular presence at a weekly breakfast event. After he broke his hip, the community “really came into focus” for him, with many neighbors offering help, checking in regularly, and picking up groceries: “And I have like 12 numbers that are my circle now on my message center. If anybody’s going to Safeway, and two people will immediately message back, ‘I’m going. I can pick something up for you’.” WESTHOME provided an ideal combination of independent living and privacy with community available whenever it was desired. As he put it: “I just go upstairs and close my door and I’m alone. But then if I open the door, I know I’m not going to be alone. Which is… I never had that before”.

#### 3.1.4. In-House Support Facilitates Access to Healthcare in LGBTQIA+ Older Adults

As well as facilitating community support, WESTHOME also provided formal services and activities, including lunches, language classes, housing advice services, case management, advocacy services, financial advice services, Tai chi, yoga, and synagogue services. Referring to the wealth of services offered, one resident said that WESTHOME “provides for everybody”, and another described it as “housing plus plus”, saying “I thought this was housing, but it’s so much more”. Some participants were aware of services and felt they didn’t need them just yet, whereas others attended events somewhat reluctantly, aware that without sufficient uptake the funding for such services might be canceled. 

The housing services coordinator received extensive positive feedback. A Black gay cisgender man in his late 50s had initially felt ambivalent about moving to WESTHOME; he was relatively happy with his previous housing and didn’t want to be “apartment rich and money poor”. However, after moving in, he was relieved to have joined such a diverse and supportive community: “Wow. I’ve used up all of my good luck for the rest of my life. Thank you”. He was one of several participants who mentioned having regular medical procedures requiring general anesthesia, meaning he needed someone to escort him home afterwards. After he mentioned this to the services coordinator in passing, she immediately arranged to help him: “To have somebody here who you can count on, in a situation like, just makes life so much easier. You might have to postpone your procedure if you can’t find anybody”.

#### 3.1.5. Mixed Feelings on LGBTQIA+-Welcoming Housing

The housing at WESTHOME was defined in several different ways—affordable, open to older adults only, and LGBTQIA+-welcoming—and residents’ responses to these markers of identity and potential vulnerability varied widely. The LGBTQIA+-welcoming nature of the building was more important to some participants than to others. For one American Indian straight transgender woman, stability was the key benefit of living at WESTHOME, much more so than the LGBTQIA+-welcoming aspect: “I feel very strongly, it’s not just the hallmark property for the LGBTQIA+ [nonprofit] thing, it’s my home because I didn’t pick it for that reason”. An American Indian/white straight transgender woman also shied away from the LGBTQIA+-welcoming nature of the building: “Well, I can only speak for myself. This space is my palace, it’s my castle, it’s my sanctuary. It’s my home. I don’t want to be identified as what the building stands for, this is my home”. For these residents, both transgender women, living in a specifically LGBTQIA+ space seemed to be incompatible with being truly at home, although they did both identify WESTHOME as their home, indicating its inclusive and supportive nature.

Other residents were very committed to WESTHOME’s LGBTQIA+-welcoming mission. One participant, an African American straight cisgender woman in her late 50s, grew up nearby but had been living in another city for 20 years. Coming home and moving to an LGBTQIA+-welcoming space meant a lot to her, although she personally identified as cisgender and heterosexual: “I mean, first of all, to be blessed to come back to the city and to be able to live in an historic building and to be around a community that I’m allied with. My children are gay and lesbian”. Being in an explicitly LGBTQIA+-welcoming environment was particularly appreciated by participants who had experienced homophobia, transphobia, and racism in previous housing. At WESTHOME, they felt able to trust others and described feeling respected, as when neighbors used their correct pronouns, for example. For others, it was important to have this community because they did not have children to look after them, had lost many friends to HIV/AIDS, or did not feel welcome in other LGBTQ spaces. For example, a resident who was a long-term HIV/AIDS survivor described how moving to WESTHOME had helped relieve some of the sadness he felt about the past, as well as the isolation he experienced due to a “lack of empathy sometimes for elder gay people” among the wider LGBTQIA+ community.

#### 3.1.6. Aging in Senior Housing

When asked about the fact that WESTHOME was for older adults only, residents reflected on the design of the apartments as well as the experience of living alongside people of similar ages. In response to Americans with Disabilities Act (ADA) adaptations such as grab rails and sliding doors, participants reported that they did not see these features as intrusive or offensive; they might have been puzzled by them at first but had to come to appreciate their presence. For example, one resident didn’t realize at first that the bathroom was larger than typical apartment bathrooms in order to accommodate a wheelchair or walker: “I was kind of like miffed that the bathroom was as big as the kitchen, because even though I’m a senior I didn’t go there just because my spirit is young, but the body tells you something else. Completely something else”.

A queer cisgender man in his late 50s, who was an architect, praised the design of the building. After moving in, he initially found it depressing to see the community room, which reminded him of his mom’s senior housing, and the rail in the bathtub. However, he had since come to value these adaptations: “It turns out that those two things are my gift. I use them all the time. I’m always in that tub, and I’m always pulling on that rail, so goes to show you that time takes its course”. However, although he was satisfied with the building, he did not feel that the policies and management were designed with older adults in mind. For example, having to use a mobile app when problems arose instead of being able to go to a full-time on-site manager for help was frustrating to him, and issues with the building’s security features made it hard for emergency services to get into the building quickly. As he put it, “That’s not really what I would call senior-friendly”.

##### Aging in Place

A white gay cisgender man who had lived in the city since 1969 and taken classes in the building in the past did not see himself being able to keep living at WESTHOME indefinitely. He reflected on his parents’ experience; they had moved to a senior complex where they could transition to different buildings with higher levels of care as needed, without having to move to a new facility altogether. He could not imagine WESTHOME being able to accommodate such a progression, saying, “It doesn’t seem to lend itself to that kind of service. I don’t think it was ever intended to become a care home”. He imagined having to go to a group home in the future: “I would be very sad to leave this building. At this point I’m resigned to it”. When another focus group member suggested that WESTHOME could facilitate in-home care as needed, his main concern was whether this might be used to justify increases in rent. He was particularly conscious of not having children or grandchildren to take care of him: “So we’re kind of on our own for the most part. That has very severe economic impact, particularly for disabled people”.

In contrast, an American Indian/white straight transgender woman felt differently about the future, picturing the community growing older together:


*It is a total gift from the universe to […] have all of my urgency taken away from me with my life. Because, as I said to my neighbors and my friends, this is the last stop for me. So I give a lot of leeway to my neighbors and my friends here, because we’re all going to be here. We’re all going to be pushing our little walkers together here.*


Being in an age-specific environment brought up different feelings for participants, including elements of ageism and ambivalence about living in senior housing, with some seeking to distance themselves from those they saw as “elderly” and others reporting that they drew solidarity and inspiration from their neighbors. After being bought out of her flat after living in the same area for 20 years, a mixed race straight transgender woman was relieved to have a permanent home where she did not have to worry about future rent increases. Stability was something she had tried hard to maintain through her adult life, after growing up in low-income housing and moving frequently, including living in hotels. She thought about the community in terms of two distinct groups, with about a quarter of the residents seeming “very, very elderly and in ill health” to her and the rest being “very outgoing […] out every day, doing things […] still working”. The sense of youth and energy at WESTHOME was uplifting for her:


*It doesn’t feel like we’re just in an elderly place, it feels like a place that’s living, growing still. That makes a big difference, because I think if it was just really sick, old, elderly people, that would zap your energy.*


For her, exercise was key to avoiding health problems associated with older age: “We’re not all walking around with walkers or… The way to avoid that is to have somewhere to do some kind of exercise”. She wanted WESTHOME to add a workout facility—the low-intensity options offered did not suit her: “I don’t like yoga, Tai Chi, I need stair master, honey”. Another resident felt that assumptions about older people had informed decisions about what amenities to provide:


*I know [LGBTQIA+ nonprofit] and [housing provider] had more of an idea that we were just going to be a lot of old people, probably not able to get out and do things, but a number of us have cars. They didn’t take that into consideration, unfortunately.*


While some residents sought to distance themselves from ageism and related stereotypes about aging, others described feeling inspired by the oldest members of the community:


*We have somebody that’s almost 100 that lives with us. We see her, and she motivates us. Every time I see her walking around this building, it encourages me to want to walk. I ain’t got to it yet but it’s good, it’s an encouragement. (African American straight cisgender woman)*


Several participants felt that living together with people of similar ages gave them a sense of solidarity and shared experiences, especially in terms of health challenges. As one put it:


*We’re all around the same age. You know what I mean? We’re all 55, 56, 57. So we’re kind of all the same. So we’re all going to be watching each other at the same rate get to the same spot.*


Similarly, a Mexican American transgender woman felt residents could understand each other’s needs and health experiences, unlike younger people: “Like, “Oh, this aches today”, or, “That aches today”, versus I can’t tell my little nephews or nieces or relate in that sense”.

## 4. Discussion

In this qualitative exploration of experiences of living in LGBTQIA+-welcoming affordable senior housing, we found that residents’ mental health and well-being was influenced by gaining housing security as well as specific features offered by WESTHOME. Once they no longer had to worry about finding or retaining stable housing, residents were able to take better care of themselves and enjoy living together with others. Living in an attractive physical environment with a strong sense of community promoted a sense of well-being and pride, and the services provided further supported residents’ health. Although most participants shared positive feelings about their surroundings, they differed in their responses to the LGBTQIA+-welcoming and senior-only nature of their new home, with some distancing themselves from these labels and others embracing them and hoping to be able to stay for the long term.

### 4.1. Identified Themes and Existing Research

#### 4.1.1. Psychological Health and Well-Being

We found that moving to stable, supportive, and affordable housing was associated with benefits for health and well-being, particularly for psychological health. With the relief of anxiety about finances and safety, residents could relax and take better care of themselves. Our findings buttress the existing literature on the positive relationship between housing stability and health. Although definitions of housing instability vary [42], there is a large and growing body of evidence highlighting associations between housing quality, cost, and location, and measures of health and healthcare utilization [42,43]. Housing stability has been associated with greater access to physical and mental healthcare, food security, improved mental health and better health outcomes for those living with HIV/AIDS [28,29].

#### 4.1.2. Discrimination and Housing Instability

Our study also aligns with other findings on the links between experiences of discrimination and housing instability [44]. We found that homophobia and transphobia were frequent contributors to the stress and insecurity experienced in previous housing situations. There is also extensive evidence on the relationship between built environments and health issues, including asthma, obesity, heart disease and mental health [45]. However, such studies typically focus on housing quality, infrastructure, safety and energy efficiency. Additionally, intersectionality in terms identifying as LGBTQIA+, an older adult, racial/ethnic minority, and other marginalized identities were reflected in this study. Intersectionality is also depicted by the numerous experiences of discrimination, violence, socioeconomic hardship, housing instability, and subsequent health disparities described by participants. The research reported in this manuscript adds qualitative insight on the interaction between visually appealing, historic physical environments, community cohesion, inclusion of LGBTQIA+ and additional minority identities, and health.

As well as appreciating the stability and appearance of their new home, participants highlighted how community, social support, and in-house services benefitted their health. Sullivan found that acceptance, safety, and inclusivity were key reasons for choosing LGBT senior living communities, and that “created community” developed surprisingly quickly, through “shared activities, care for one another, and the shared connection of being sexual minority seniors” (p. 241) [5]. The residents of WESTHOME were in the unusual position of being the first to move into their housing, meaning they had a unique opportunity to form a new sense of community. It would be valuable to explore how this community develops and changes over time, as some original residents leave and new people join the community.

#### 4.1.3. Community and Psychological Health

The strong link between community and psychological health for the residents of WESTHOME also echoes a qualitative study on aging preferences of older lesbians [35], in which participants identified social support and community as important for health. In our study, formal service provision was important in facilitating access to healthcare services and building close bonds between residents. Previous studies on supportive services in (non-LGBTQIA+-specific) senior housing describe self-reported benefits for health, reduced use of inpatient services, and reduced growth rates for Medicare expenditures [46,47,48]. However, researchers have documented increases in short-term costs associated with improved healthcare access, with reduced longer-term expenditures due to effective prevention and health maintenance [49]. Further research using longitudinal methods and medical record data could help to shed more light on the health and healthcare implications of supportive housing for LGBTQIA+ older adults over time.

#### 4.1.4. Mixed Feelings about Ageism and LGBTQIA+-Welcoming Senior Housing

Finally, we found that the LGBTQIA+-welcoming, older adult-specific, affordable nature of this housing evoked mixed feelings. Existing evidence is inconsistent on the benefits and drawbacks associated with age-segregated housing in terms of emotional well-being [50,51]). In a qualitative study, researchers found that resistance to age-segregated settings may be driven by preferences for socializing with younger people and for a greater degree of independence [39]. Benefits of age-segregated settings reported in their study included companionship and community, safety and security, and affordability, although the authors raise concerns about the extent to which older adults are forced into age-segregated settings due to a lack of alternatives. Our study found that some residents expressed particular resistance to the LGBTQIA+-segregated nature of WESTHOME. This novel finding may reflect internalization of the homophobia and transphobia that many participants experience throughout their life; in comparison, ageism was a more recent challenge, and fewer participants expressed concerns about age-segregation.

It is also important to note that the lottery was centrally administered alongside the other affordable housing available in the city, in the context of huge demand for housing and increasing rent costs. Lottery entry was open to anyone who met the age and income eligibility requirements, and participants had applied for countless housing lotteries in the past without success. For some, WESTHOME was one of many options they would have accepted, rather than an environment that they had specifically and actively sought out for its LGBTQIA+-welcoming characteristics. It is important for those developing similar housing developments around the U.S. to keep in mind these structural constraints and their potential impact on community building.

### 4.2. Strengths and Limitations

This is the first study to investigate experiences of residents moving to an LGBTQIA+-welcoming affordable senior housing development. We designed and followed a robust analytic plan, and our sample included a large proportion of the total number of residents in the building. However, we also acknowledge the limitations of this study, which was undertaken at a single site, less than a year after residents moved in and perhaps during a “honeymoon” period. We conducted focus groups on site, and recruitment was facilitated by in-house staff. There is a potential for social desirability bias given that participants may have felt pressured to give positive feedback, and being among neighbors in a focus group could have made some reluctant to disagree or express dissatisfaction. Finally, not all participants identified as LGBTQIA+, although all were living in an LGBTQIA+-welcoming community. The findings of this study may not be reflective of the experiences of all residents of LGBTQIA+-welcoming housing, but they may provide important insights for those planning and evaluating similar housing across the U.S. There is also a need for future research on whether the desire and need for LGBTQIA+ senior housing is specific to certain generations and geographic regions.

## 5. Conclusions

LGBTQIA+ older adults face unique challenges in obtaining affordable and stable housing, as well as greater health concerns compared to their cisgender heterosexual peers. Affordable, supportive, and inclusive housing for LGBTQIA+ older adults may provide benefits for physical, mental, and social health by fostering community and facilitating access to services. This research forms the basis of a larger longitudinal mixed-methods study of the effects of affordable and inclusive housing on long-term health and healthcare costs, which will enhance our understanding of the time course and trajectories of these effects. As the LGBTQIA+ older adult population continues to grow, it is essential to provide appropriate and accessible housing solutions to support health and well-being of LGBTQIA+ populations throughout older age, recognizing their unique needs and preferences. Public health implications include: informing housing agencies and advocates as well as public health and clinical professionals of the need to improve programs and services; increasing understanding of the housing and health needs of LGBTQIA+ older adults; and initial evidence on the potential benefits of LGBTQIA+ age-friendly housing on the physical and mental health of LGBTQIA+ older adults, including the role that housing can play in increasing access and utilization of healthcare and social services.

## Figures and Tables

**Table 1 ijerph-19-01699-t001:** Demographic characteristics of sample (*n* = 21).

Characteristic	Percent
Age mean (sd)	61 (5.6)
Hispanic/Latinx	14%
Race/ethnicity	
American Indian/Alaska Native	14%
Black/African American	38%
Mexican	5%
Mixed heritage	5%
White	48%
Another	5%
Highest education completed	
≤High school	29%
Some college/technical training	24%
2-year college degree	5%
4-year college degree	19%
Masters/Professional degree	24%
Sex assigned at birth	
Female	24%
Male	76%
Gender identity	
Female	33%
Male	57%
Transgender female	14%
Sexual orientation	
Asexual	10%
Bisexual	10%
Gay/homosexual	34%
Heterosexual/straight	40%
Queer	14%
Another identity	5%

## Data Availability

The data presented in this study are available on request from the corresponding author. The data are not publicly available to protect confidentiality of the research participants.

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
