# Peer review of "Building H.O.U.S.E (Healthy Outcomes Using a Supportive Environment): Exploring the Role of Affordable and Inclusive Housing for LGBTQIA+ Older Adults"

_ijerph, 2022, doi:10.3390/ijerph19031699_

Round 1

Reviewer 1 Report

Abstract

The abstract is too vague.  Please state the aims of the study or the research question specifically.

Introduction

Some demographic statistics on the LGBTQ+ community where this housing development is located would be helpful.

What theoretical framework did the authors use to conceptualize this study?
When discussing health and income disparities, it would also be helpful to discuss resiliencies.  Perhaps because of discrimination and heterosexism, LGBTQ+ individuals have formed strong supportive networks outside of their families.  What role does that play in housing choices?

Is living in exclusively or primarily LGBTQ+ housing welcomes by most members of this community or just a specific generation?

Is it geographically specific to where this housing is located?  Could it be different in large urban cities with a sizable LGBTQ+ population?

Methods

Was IRB approval received? 

Who developed the focus group questions?

How many participants in each focus group?  How long did they last?

Findings

Very well organized and explained.

It would be helpful to have a section on what challenges they faced or any negative consequences.  Its interspersed throughout the discussion and gets lost in narrative.  What are the downsides if any to this situation?

Discussion

I think the authors need to separate mental and emotional health from physical.  Since this is not a RCT, impact on physical health is not clear.  But emotional health self reporting is clear and needs to be highlighted.

I would separate this section with sub-headings and also link it back to your literature review.

Implications

What are the implications from these findings for practice and policy with LGBTQ+ older adults?

Author Response

Abstract

The abstract is too vague.  Please state the aims of the study or the research question specifically.

Response: Thank you. We now include the following in the abstract to reflect the study aims. "Focus groups interviews were conducted with 21 older adults to explore the lived experiences and potential health benefits of living in a new LGBTQIA+ welcoming senior housing."

Introduction

Some demographic statistics on the LGBTQ+ community where this housing development is located would be helpful.

Response: Thank you for this suggestion. We now include a figure of the number of LGBTQIA+ older adults in the Materials and Methods on page 3 lines 115-117. "In this region, it is estimated that between 18,000 and 20,000 LGBTQIA+ older adults aged 60 years or older currently reside in Western U.S. Metropolitan area [7]." 

What theoretical framework did the authors use to conceptualize this study?

Response: Thank you for pointing out. We now mention the theoretical framework used under the Materials and Methods on page 3 lines 138-140. "The theoretical framework for this study was the Health Equity Promotion, provides a life course perspective on how historical and social factors have influenced the health of LGBTQIA+ older adults [42]."

When discussing health and income disparities, it would also be helpful to discuss resiliencies.  Perhaps because of discrimination and heterosexism, LGBTQ+ individuals have formed strong supportive networks outside of their families.  What role does that play in housing choices?

Response: Thank you for this important point. We included this under our section on health disparities on page 2, lines 74-78. "However, potential resilience factors should also be considered for LGBTQIA+ older adults, such as community connections, unique social networks including chosen family and friends, and greater engagement in social, leisure and wellness activities [1]. These factors may also be important to linkages to housing and related health resources."

Is living in exclusively or primarily LGBTQ+ housing welcomes by most members of this community or just a specific generation? Is it geographically specific to where this housing is located?  Could it be different in large urban cities with a sizable LGBTQ+ population?

Response: These are very important importants that were not well addressed by our study. We added these points to our section on strengths and limitations on page 13 lines 580-581. "There is also a need for future research on whether the desire and need for LGBTQIA+ senior housing is specific to certain generations and geographic regions."

Methods

Was IRB approval received? 

Response: Yes, this study received IRB approval. We report this on page 13 under the Institutional Review Board Statement and under Materials and Methods on page 3 lines 119-120.

Who developed the focus group questions?

Response: We added this to the methods and materials on page 4, line 154 that the focus group guide was developed by the authors. 

How many participants in each focus group?  How long did they last?

Response: We mentioned on page 4, lines 177-179 that 21 participants were included with the breakdown by focus group and we have now added that interviews lasted between 75 to 92 minutes.

Findings

Very well organized and explained.

Response: Thank you

It would be helpful to have a section on what challenges they faced or any negative consequences.  Its interspersed throughout the discussion and gets lost in narrative.  What are the downsides if any to this situation?

Response: Downsides/challenges were not a major theme that emerged from our qualitative analysis of the focus group transcripts, so instead we tried to incorporate both the positive and negative aspects under each theme. We hope that the reviewers will accept this balance of both positive and negative experiences under each of these core themes.

Discussion

I think the authors need to separate mental and emotional health from physical.  Since this is not a RCT, impact on physical health is not clear.  But emotional health self reporting is clear and needs to be highlighted.

Response: Thank you for this suggestion. We have revised the discussion to emphasize the emotional/psychological health benefits rather than broad health.

I would separate this section with sub-headings and also link it back to your literature review.

Response: Thank you. We have now added heading and worked to ensure the literature review was highlighted in our discussion.

Implications

What are the implications from these findings for practice and policy with LGBTQ+ older adults?

Response: Thank you for this recommendation. We now add the following to our conclusion paragraph on page 13 lines 600-605. "Public health implications include: informing both housing agencies and advocates as well as public health and clinical professionals to improve programs and services; increasing understanding of the housing and health needs of LGBTQIA+ older adults; and initial evidence on the potential benefits of LGBTQIA+ age-friendly housing on the physical and mental health of LGBTQIA+ older adults, including the role that housing can play in increasing access and utilization of healthcare and social services."

Reviewer 2 Report

This is a very well-written paper that describes a novel study which addresses the increasingly recognized importance of adequate, affordable, supportive housing. The unique situation of LGBTQIA+ aging adults with regard to housing and health is presented convincingly.  Congratulations on an excellent paper. I have only a few suggestions/questions:

  1. Were there more than 21 people who volunteered to be in the study?  If so, how did you decide and communicate who would be included?  
  2. The ageism reflected in ambivalence about living in senior housing could be named as such in the results. It is mentioned in the discussion, but there were some notable ageist sentiments in the findings.
  3. Given the work we still have to do on understanding and addressing intersectionality, and given the fact that this paper does implicitly speak to that topic, it would be a contribution to that literature if the authors would spend a few sentences talking about it. I understand the potential disadvantages of making brief reference to the topic, and I trust the authors to make the right decision about it.

Author Response

  1. Were there more than 21 people who volunteered to be in the study?  If so, how did you decide and communicate who would be included?  

Response: Thank you for this suggestion. We now add a line on page 3 under methods and materials, line 145-146 that 26 participants were recruited initially. "Twenty-six participants were initially recruited to participate and contacted by phone to schedule the date of the focus group interview."

2. The ageism reflected in ambivalence about living in senior housing could be named as such in the results. It is mentioned in the discussion, but there were some notable ageist sentiments in the findings.

Response: Thank you for this suggestion. We have now incorporated language on ageism in the results and discussion. 

3. Given the work we still have to do on understanding and addressing intersectionality, and given the fact that this paper does implicitly speak to that topic, it would be a contribution to that literature if the authors would spend a few sentences talking about it. I understand the potential disadvantages of making brief reference to the topic, and I trust the authors to make the right decision about it.

Response: Thank you for this suggestion. We have added a few lines under section 4.1.2 Discrimination and housing instability on page 12, lines 527-530 to highlight this intersectional lense.  "Additionally, intersectionality in terms identifying as LGBTQIA+, an older adult, racial/ethnic minority, and other marginalized identities were reflected in this study. Internationality is also depicted by the numerous experiences of discrimination, violence, socioeconomic hardships, housing instability, and subsequent health disparities described by participants."

Round 2

Reviewer 1 Report

The authors have made substantive changes to the manuscript that has significantly strengthened it.  I suggest it be accepted.